# Comparative Evaluation of Modified Vaccinia Ankara as a Surrogate Virus for Disinfectant Efficacy Testing Against AIV, FMDV, and ASFV

**DOI:** 10.3390/v17091156

**Published:** 2025-08-23

**Authors:** Sok Song, Su-Jeong Kim, Kyu-Sik Shin, So-Hee Park, Yong Yi Joo, Bokhee Han, Cho-Yeon Lee, Gong-Woo Park, Hyun-Ok Ku, Wooseog Jeong, Choi-Kyu Park

**Affiliations:** 1Veterinary Drugs & Biologics Division, Animal and Plant Quarantine Agency, Gimcheon 39660, Republic of Korea; ssoboro@korea.kr (S.S.); kimsujeong27@korea.kr (S.-J.K.); soso9354@naver.com (K.-S.S.); sh0526@korea.kr (S.-H.P.); wndyddl7091@korea.kr (Y.Y.J.); joy2996@korea.kr (B.H.); sol77lee@korea.kr (C.-Y.L.); pgongwoo@naver.com (G.-W.P.); kuho@korea.kr (H.-O.K.); 2Institute for Veterinary Biomedical Science, College of Veterinary Medicine, Kyungpook National University, Daegu 41566, Republic of Korea

**Keywords:** surrogate virus, virucidal efficacy testing, MVA, ASFV, AIV, FMDV

## Abstract

Surrogate viruses provide a safe and scalable alternative for evaluating disinfectant efficacy when access to high-risk pathogens is restricted. This study evaluated the potential of Modified Vaccinia Ankara (MVA) virus, which can be handled under BSL-1/2 conditions, as a surrogate for avian influenza virus (AIV), foot-and-mouth disease virus (FMDV), and African swine fever virus (ASFV). A total of 64 commercially available disinfectants—classified into four major chemical groups: quaternary ammonium compounds, oxidizing agents, PPMS-based formulations, and organic acids—were tested in suspension assays using a ≥4 log reduction as the efficacy criterion. MVA showed the strongest predictive performance for FMDV (r = 0.671, AUC = 0.83), supporting its use for both binary classification and approximate quantitative prediction. Although its correlation with ASFV was weaker (r = 0.175), the classification performance remained moderate (AUC = 0.78), indicating conditional applicability. While MVA exhibited no meaningful correlation with AIV, its higher chemical resistance meant that disinfectants effective against MVA were consistently effective against AIV. These results support the use of MVA as a conservative exclusion tool for fragile viruses. Overall, the findings demonstrate that MVA can serve as a practical surrogate virus for disinfectant efficacy testing against FMDV, ASFV, and AIV, with application strategies tailored to each virus’s characteristics.

## 1. Introduction

Transboundary animal diseases, such as African swine fever (ASF), foot-and-mouth disease (FMD), and avian influenza (AI), pose major threats to global livestock production due to their high transmissibility and severe economic impacts [1,2,3]. These diseases are caused by African swine fever virus (ASFV), foot-and-mouth disease virus (FMDV), and avian influenza virus (AIV), respectively. ASFV, FMDV, and AIV are globally significant transboundary pathogens with varying degrees of endemicity and pandemic potential [4,5,6,7,8,9]. Regardless of vaccine availability, disinfection remains essential for controlling FMD, AI, and especially ASF, for which no widely available vaccine exists [10,11,12].

The physicochemical properties of these viruses differ markedly. ASFV, a large enveloped DNA virus with a multilayered icosahedral structure, exhibits strong environmental stability and resistance to chemical agents [13,14,15]. AIV, also enveloped but structurally more fragile, is generally more susceptible to inactivation [16,17]. In contrast, FMDV is a small non-enveloped RNA virus that is typically more resistant to chemical disinfectants than enveloped viruses [18,19]. These structural differences, together with variability in vaccine effectiveness, underscore the need for disinfection strategies tailored to each virus.

Despite these differences, field-level disinfection often employs a unified approach. In the Republic of Korea, centralized facilities frequently disinfect vehicles from swine, poultry, and cattle farms using a single product at the highest concentration required for any target pathogen. In such contexts, operational simplicity and broad-spectrum efficacy take precedence over pathogen-specific protocols. This highlights the need for a single surrogate virus that can conservatively represent multiple high-consequence pathogens. If the surrogate is more resistant than the target viruses, its inactivation threshold can serve as a robust benchmark for ensuring efficacy across all three. Evaluating the suitability of a surrogate virus across structurally diverse pathogens is therefore critical for both regulatory assurance and practical disinfection reliability.

The need for a unified surrogate virus is further emphasized by regulatory and biosafety constraints associated with live-virus testing. In the Republic of Korea, the Animal and Plant Quarantine Agency (APQA) oversees the approval and post-market surveillance of veterinary disinfectants. To ensure product reliability, APQA annually retests approximately 80 disinfectants out of more than 200 registered formulations. Virucidal efficacy is typically evaluated using live-virus suspension tests [20,21]. However, for high-risk pathogens such as ASFV, FMDV, and AIV, these tests require biosafety level 3 (BSL-3) containment, limiting accessibility and scalability. Most manufacturers lack such facilities and must rely on certified third-party laboratories. This reliance poses a global challenge, reinforcing the need for safe, scalable alternatives to live-virus testing. To address this, several regulatory bodies—including the U.S. Environmental Protection Agency (EPA) and the European Committee for Standardization (CEN/TC 216)—permit the use of surrogate viruses, selected based on biosafety, stability, and structural similarity to target viruses [20,22,23]

Modified Vaccinia Ankara (MVA) virus, an attenuated poxvirus, is widely used as a surrogate for enveloped viruses because of its robust stability, multilayered envelope, and favorable biosafety profile [24,25]. MVA is currently accepted in various human and veterinary disinfectant testing protocols [21,26,27]. However, its broader utility—particularly for non-enveloped viruses, such as FMDV—remains uncertain. In Europe, for example, regulatory guidelines mandate the use of bovine enterovirus type 1 (ECBO) to represent non-enveloped viruses such as FMDV [28,29]. Given the divergent properties of MVA and ECBO, the predictive reliability of MVA across different virus types warrants further investigation [30,31].

Historically, surrogate virus selection has relied on the assumption that structural similarity corresponds to similar susceptibility [32]. However, comprehensive, data-driven evaluations across multiple virus–disinfectant combinations remain limited. Previous Korean studies have examined candidate surrogates such as MVA, MS2, and ECBO, but these were largely descriptive and focused on active ingredients [18,25].

This study quantitatively evaluates the predictive suitability of MVA as a surrogate virus for disinfectant efficacy testing against AIV, FMDV, and ASFV. Sixty-four commercially available disinfectants representing four major chemical classes were tested for their inactivation efficacy against all four viruses. Correlation analysis, receiver operating characteristic (ROC) curves, and group-wise performance metrics were applied to assess MVA’s surrogate potential. The results are expected to inform surrogate virus selection strategies and contribute to the development of scalable, risk-adjusted disinfectant evaluation systems for high-consequence animal diseases.

## 2. Materials and Methods

### 2.1. Viruses and Cells

Three representative animal viruses and MVA were used in this study: ASFV (ASFV-PJ-VaCIn1; KVCC-VR2400015, Korea Veterinary Culture Collection, Gimcheon, Republic of Korea), FMDV vaccine strain of serotype O (O/SKR/Boeun/2017; KVCC-VR1700004, Korea Veterinary Culture Collection, Gimcheon, Republic of Korea), AIV (H9N2 strain A/chicken/Korea/MS96/1996; KVCC-VR1100013, Korea Veterinary Culture Collection, Gimcheon, Republic of Korea), and MVA (ATCC^®^ VR-1566™, Manassas, VA, USA).

ASFV, FMDV, and MVA were propagated in Vero cells (ATCC^®^ CCL-81™, Manassas, VA, USA), LFBK cells (RRID: CVCL_RX26; derived from porcine fetal kidney; KVCC, Korea Veterinary Culture Collection, Gimcheon, Republic of Korea), and BHK-21 cells (ATCC^®^ CCL-10™, Manassas, VA, USA), respectively. All cell lines were maintained in Dulbecco’s Modified Eagle’s Medium (DMEM) or Minimum Essential Medium (MEM) (Corning Inc., Corning^®^, NY, USA), supplemented with 5–10% fetal bovine serum (FBS, Gibco, Thermo Fisher Scientific, Waltham, MA, USA), Antibiotic-Antimycotic (100 U/mL penicillin, 100 µg/mL streptomycin, and 0.25 µg/mL amphotericin B; Gibco, Thermo Fisher Scientific, Waltham, MA, USA), and 200 mM L-glutamine (Gibco, Thermo Fisher Scientific, Waltham, MA, USA), at 37 °C in a 5% CO_2_ incubator (Forma™ Direct Heat CO_2_ Incubator, Thermo Fisher Scientific, Waltham, MA, USA).

Viral stocks were prepared by infecting confluent monolayers, harvesting the supernatants upon the appearance of cytopathic effects (CPE), and clarifying them by centrifugation at 3000× *g* for 10 min. Virus titers were determined in triplicate using the 50% tissue culture infectious dose (TCID_50_) method and calculated by the Spearman–Kärber formula. Typical titers ranged from 10^6^ to 10^8^ TCID_50_/mL.

AIV was propagated in the allantoic cavities of 10-day-old specific pathogen-free (SPF) embryonated chicken eggs. After 72 h of incubation at 37 °C, the allantoic fluid was harvested, clarified by centrifugation at approximately 3000× *g* for 10 min, and stored at –80 °C. The presence of virus was confirmed by hemagglutination assay, and titers were expressed as the 50% egg infectious dose (EID_50_/mL).

### 2.2. Disinfectants

In 2024, 64 disinfectants were selected from approximately 80 products collected by APQA for post-market surveillance, based on label claims indicating efficacy against AIV, FMDV, and ASFV. The selected disinfectants were classified into four major chemical classes commonly used in veterinary disinfection:(1)Quaternary ammonium compounds (QACs; *n* = 22);(2)Oxidizing agents (*n* = 8);(3)Potassium peroxymonosulfate (PPMS)-based products (*n* = 28);(4)Organic acids (*n* = 6).

Table 1 summarizes the classification, number of products, and representative formulations for each category. All disinfectants were diluted in sterile distilled water according to either the manufacturers’ instructions or standard use concentrations recommended for veterinary applications. Working solutions were freshly prepared immediately before testing and used within 30 min of dilution to ensure chemical stability and consistent performance.

### 2.3. Virucidal Efficacy Test

The virucidal efficacy assay was conducted using the quantitative suspension method in accordance with guidelines established by the APQA, Republic of Korea. Each disinfectant dilution, prepared in hard water (pH 7.0 ± 0.2; CureBio, Seoul, Republic of Korea), was mixed with an equal volume of virus suspension (1:1) and incubated at 4 °C for 30 min. To maintain a homogeneous reaction, samples were briefly vortexed every 10 min during incubation. Following exposure, the mixtures were neutralized by adding an equal volume of DMEM supplemented with 10% FBS. Neutralized solutions were serially diluted tenfold in hard water and inoculated at 100 µL per well onto confluent monolayers in 96-well microplates: BHK-21 cells for MVA, Vero cells for ASFV, and LFBK cells for FMDV. The plates were incubated at 37 °C with 5% CO_2_ for 3–6 days, depending on the virus. CPEs were monitored daily using an inverted microscope. Samples that showed cytotoxic effects on cell cultures were excluded from efficacy calculation. Viral titers were calculated by the Spearman–Kärber method and expressed as log_10_ TCID_50_/mL. Virucidal efficacy was determined as the log_10_ reduction in viral titer compared to an untreated virus control processed in parallel.

For AIV, virucidal efficacy was assessed using a modified suspension test followed by embryonated egg inoculation. After disinfectant treatment and neutralization, 200 µL of each neutralized sample was inoculated into the allantoic cavity of 10-day-old SPF embryonated chicken eggs (five eggs per sample). Eggs were incubated at 37 °C and monitored daily for embryonic death over a 5-day period. Allantoic fluid was harvested from surviving embryos and tested for the presence of AIV by hemagglutination (HA) assay using 0.5% chicken red blood cells. Eggs that showed nonspecific embryo mortality due to disinfectant toxicity were excluded from the final assessment. A disinfectant was considered effective if no HA activity was detected in any of the inoculated eggs, indicating complete viral inactivation. All experiments were performed independently in triplicates to ensure reproducibility.

### 2.4. Data Analysis

All statistical analyses were performed using GraphPad Prism (version 8.0.2, GraphPad Software, San Diego, CA, USA) and Microsoft Excel (Microsoft Corporation, Redmond, WA, USA). Virucidal efficacy was expressed as the log_10_ reduction in virus titer relative to untreated controls, with a reduction of ≥4 log_10_ defined as the threshold for effective inactivation. Pearson correlation coefficients (r) were calculated to evaluate linear relationships between MVA and each target virus (ASFV, FMDV, and AIV). Statistical significance was assessed using two-tailed p-values (α = 0.05) and 95% confidence intervals. Receiver operating characteristic (ROC) curves were generated from binary classification outcomes (effective vs. ineffective, based on the ≥4 log_10_ reduction threshold), and the area under the curve (AUC) was calculated using the trapezoidal rule. When ROC curves could not be generated due to uniform class labels, the AUC was reported as not calculable (NaN). Descriptive statistics, including the mean, standard deviation (SD), and coefficient of variation (CV), were used to summarize log_10_ reduction values within each disinfectant chemical group. All statistical tests were conducted at a significance level of 0.05.

## 3. Results

### 3.1. Comparative Virucidal Efficacy of Disinfectants Against Four Viruses

To visualize patterns of virucidal efficacy across chemical classes, a heatmap was generated using the log_10_ reduction values of 64 commercial disinfectants tested against AIV, FMDV, ASFV, and the surrogate virus MVA (Figure 1).

Across all chemical groups, most disinfectants demonstrated high efficacy against AIV and FMDV, with mean log_10_ reduction values generally exceeding 4. In contrast, efficacy against ASFV—and particularly against MVA—was more variable and tended to be lower. A notable proportion of products, especially those in the QAC- and acid-based groups, achieved ≤ 3 log_10_ reduction against MVA.

Among QAC-based disinfectants (*n* = 22), strong and consistent activity was observed against AIV and FMDV. However, reductions for ASFV and MVA varied markedly among products, indicating weaker and less uniform activity. Notably, several QAC products that achieved >5 log_10_ reduction against ASFV exhibited poor performance (<3 log_10_) against MVA, suggesting discrepancies between target virus and surrogate inactivation profiles.

Oxidizing agents (*n* = 8) exhibited moderate activity across all viruses, with particularly low efficacy against MVA. None of the products consistently achieved ≥5 log_10_ reduction across all four viruses, underscoring the challenge of attaining broad-spectrum performance within this chemical group.

In the PPMS group (*n* = 28), many products demonstrated broad-spectrum efficacy, with consistent reductions against AIV, FMDV, and ASFV. However, reductions against MVA were frequently lower, indicating that even high-performing disinfectants may have limited activity against the surrogate under the test conditions.

Organic acid-based products (*n* = 6) generally showed moderate efficacy against AIV, FMDV, and ASFV. In contrast, MVA displayed lower susceptibility, with most products failing to achieve >4 log_10_ reduction, although a few exceeded this threshold. While the small sample size limits definitive conclusions, these findings suggest greater resistance of MVA to organic acids under the tested conditions.

Overall, the heatmap analysis revealed distinct efficacy profiles across chemical groups and substantial variation in MVA susceptibility. These patterns indicate that surrogate virus data should be interpreted in the context of the chemical mechanisms of action for each disinfectant group.

### 3.2. Correlation of Virucidal Efficacy Between MVA and Target Viruses

To evaluate the potential of MVA as a surrogate virus, pairwise correlations of log_10_ reduction values were analyzed among MVA, ASFV, FMDV, and AIV across 64 commercially available disinfectants. The resulting scatter plot matrix (Figure 2) was designed to visualize both quantitative and distributional relationships. Diagonal panels display histograms of each virus’s reduction values, while the off-diagonal panels present pairwise scatter plots with fitted linear regression lines and 95% confidence intervals.

Histogram analysis showed that AIV and ASFV responses were tightly clustered above the 4 log_10_ threshold, indicating consistently high virucidal efficacy across disinfectants. In contrast, MVA and FMDV displayed broader and more variable distributions, spanning values below and above the threshold. MVA exhibited the widest spread, reflecting considerable heterogeneity in susceptibility across different formulations.

Scatter plot analysis revealed the strongest correlation between MVA and FMDV (r = 0.671), with data points closely aligned to the regression line and narrow confidence bounds, supporting a robust linear relationship. A weak positive correlation was observed between MVA and ASFV (r = 0.175), but the wide dispersion of data points suggests low predictive reliability. The correlation between MVA and AIV was negligible (r = 0.002), with an almost flat regression slope and broad confidence interval, indicating no meaningful association.

Among the target viruses, only ASFV and FMDV showed a weak correlation (r = 0.219), while correlations between ASFV and AIV (r = 0.079) and between FMDV and AIV (r = 0.032) were minimal. These results suggest limited overlap in disinfectant susceptibility profiles among the target viruses and highlight their low cross-predictive potential.

Overall, the statistical and visual evidence in Figure 2 supports the partial use of MVA as a surrogate for FMDV efficacy testing under suspension test conditions. However, its applicability to ASFV appears limited, and its use for predicting AIV inactivation is not supported.

### 3.3. Classification Performance of MVA for Predicting Virucidal Efficacy Against Target Viruses

To further evaluate the predictive validity of MVA, receiver operating characteristic (ROC) curve analysis was conducted to assess its ability to identify high-efficacy disinfectants (≥4 log_10_ reduction) for each target virus (Figure 3). ROC analysis plots the true positive rate against the false positive rate across varying threshold settings, with the area under the curve (AUC) quantifying overall performance. An AUC closer to 1.0 indicates better discriminative ability, whereas a value near 0.5 suggests performance equivalent to random chance.

MVA demonstrated good discriminative performance for FMDV and ASFV, with AUC values of 0.83 and 0.78, respectively. These results indicate that MVA can reasonably classify disinfectants effective against FMDV and ASFV, with clear separation between true positives and false positives. In contrast, the ROC curve for AIV was not interpretable (AUC = NaN), as nearly all disinfectants achieved ≥4 log_10_ reduction against AIV, resulting in no meaningful variability. This extreme class imbalance prevented the model from distinguishing between effective and ineffective outcomes.

To complement the ROC-based evaluation, additional classification metrics derived from MVA predictions were assessed, including sensitivity, specificity, accuracy, positive predictive value (PPV), and negative predictive value (NPV) (Figure 4). For FMDV, MVA achieved a sensitivity of 0.91, indicating that most effective disinfectants were correctly identified, and a specificity of 0.71, reflecting a moderate ability to exclude ineffective products. Overall accuracy reached 0.84, with PPV and NPV of 0.87 and 0.79, respectively, suggesting reliable performance with relatively low false-positive and false-negative rates.

For AIV, MVA achieved perfect classification in this dataset, with sensitivity, PPV, and accuracy all reaching 1.00. This indicates that all disinfectants predicted to be effective by MVA were indeed effective against AIV. However, as no disinfectants were classified as ineffective (TN = 0), specificity and NPV could not be calculated. This limitation reflects characteristics of the dataset and warrants cautious interpretation.

For ASFV, MVA achieved a sensitivity of 0.83, indicating good detection of effective disinfectants. Specificity was lower at 0.64, reflecting a moderate rate of false positives. Accuracy was 0.80, and PPV was high at 0.92, showing strong predictive power for classifying disinfectants as effective. However, the relatively low NPV of 0.44 suggests that ineffective disinfectants were less reliably identified, potentially leading to overestimation of efficacy in borderline cases.

### 3.4. Comparison of Disinfectant Efficacy by Chemical Group

To examine whether virucidal efficacy varied by chemical class, the log_10_ reduction values of 64 disinfectants were grouped into four categories—QAC-based, oxidizers, PPMS-based, and acids—and compared across the four tested viruses (AIV, FMDV, ASFV, and MVA) (Figure 5, Table 2).

For AIV, all chemical groups exhibited high virucidal efficacy, with mean log_10_ reductions ranging from 4.92 (acids) to 5.21 (oxidizers and PPMS-based). The coefficient of variation (CV) was low across all groups (7.7–12.2%), indicating consistent performance.

FMDV reduction values were moderately lower than those for AIV. Oxidizers and PPMS-based products again showed higher efficacy (mean log_10_ reductions > 4.5), whereas QAC-based and acid-based groups showed comparatively lower reductions (3.85 and 4.11, respectively). CVs were low to moderate, with oxidizers showing the least variability (4.6%).

For ASFV, PPMS-based and oxidizer disinfectants achieved the highest mean reductions (4.74 and 4.73 log_10_, respectively), while QAC-based products showed greater variability (CV = 20.5%). Acid-based disinfectants demonstrated consistent performance (CV = 6.3%) despite slightly lower mean reductions (4.57 log_10_).

MVA exhibited the greatest resistance among the tested viruses. QAC-based products performed poorly (mean = 2.37 log_10_ reduction, CV = 31.6%), whereas oxidizers and PPMS-based disinfectants were more effective (means = 3.91 and 4.06 log_10_, respectively). Acid-based products showed moderate efficacy with high variability (mean = 3.58, CV = 29.6%).

Overall, oxidizers and PPMS-based disinfectants demonstrated the most consistent and broadly effective performance across all virus types, including more resistant targets such as ASFV and MVA. In contrast, QAC-based products showed variable efficacy and poor performance against MVA, indicating limitations for structurally robust enveloped viruses. Acid-based products generally displayed moderate to high efficacy, depending on the target virus.

Overall, oxidizers and PPMS-based disinfectants demonstrated the most consistent and broadly effective performance across all virus types, including AIV, and were particularly effective against more resistant targets such as ASFV and MVA. In contrast, QACs-based products exhibited variable efficacy and poor performance against MVA, suggesting limitations in their use against structurally robust enveloped viruses. Acid-based products generally showed moderate to high efficacy, depending on the target virus.

## 4. Discussion

For quarantine purposes, this study evaluated the utility of MVA as a surrogate virus for assessing the virucidal efficacy of 64 commercial disinfectants, based on the highest effective concentrations against AIV, FMDV, and ASFV. MVA showed the strongest predictive alignment with FMDV, with a moderate correlation (r = 0.671) and good ROC performance (AUC = 0.83). For ASFV, the correlation was weak (r = 0.175), but the AUC value of 0.78 indicates some discriminatory capacity. This relatively low correlation suggests a risk of false-positive predictions when relying solely on MVA, and confirmatory testing with the target virus may be necessary in borderline cases or regulatory decision-making contexts.

MVA showed negligible correlation with AIV (r = 0.002), and ROC analysis was not applicable due to uniformly high inactivation rates. Although predictive modeling was not feasible, MVA’s high resistance means that any disinfectant effective against MVA is also effective against AIV, supporting its role as a conservative exclusion tool. As summarized in Table 3, virucidal efficacy against MVA served as a conservative indicator for AIV, FMDV, and, to a lesser extent, ASFV. MVA’s multilayered envelope and complex dsDNA structure likely contributes to its resistance to various disinfectant-induced damages, supporting its use as a robust benchmark virus [24,26,33]. These features may explain its ability to withstand a broad range of disinfectant chemistries.

For FMDV, the observed correlation with MVA and alignment of inactivation thresholds support MVA’s use not only for binary classification but also for approximate quantitative prediction. AIV’s high susceptibility limits predictive modeling, but MVA’s higher resistance can reliably exclude ineffective products in regulatory applications. The comparison between FMDV and MVA demonstrates that viruses with distinct structural features—FMDV being a non-enveloped RNA virus and MVA a large enveloped DNA virus—can still exhibit similar resistance profiles [34,35,36]. Conversely, ASFV and MVA, both large and structurally complex dsDNA viruses, showed only weak correlation, indicating that even pronounced structural complexity does not guarantee predictive alignment [25,37]. This discrepancy may be attributable to differences in envelope composition and architecture. ASFV possesses an internal lipid membrane enclosed by an icosahedral capsid and an additional outer lipid envelope, whereas MVA exhibits a multilayered envelope with distinct protein–lipid arrangements and embedded membrane proteins. Such differences may influence susceptibility to specific chemical classes of disinfectants, leading to divergent inactivation patterns despite overall structural complexity. Therefore, surrogate virus selection should consider not only the presence or absence of an envelope or overall structural complexity, but also empirically validated patterns of disinfectant susceptibility [38].

Chemical class also significantly influenced virucidal outcomes [39,40]. Oxidizing agents and PPMS-based disinfectants displayed broad-spectrum efficacy, likely due to their capacity to damage viral envelopes, denature proteins, and oxidize nucleic acids [41,42,43]. In contrast, QACs were highly effective against AIV and FMDV but less so against structurally complex viruses like ASFV and MVA. Although organic acid-based disinfectants showed variable performance, the small sample size within this category limits the generalizability of the findings.

Taken together, these findings support the selective use of MVA as a conservative surrogate virus in disinfectant efficacy testing, particularly for FMDV and, to a lesser extent, ASFV. In settings lacking high-containment facilities, MVA offers a biosafety-compatible and predictive alternative for preliminary screening of disinfectant efficacy against resilient viruses. For fragile viruses such as AIV, MVA offers value as a conservative screening tool to ensure baseline efficacy.

## 5. Conclusions

This study supports the selective use of MVA as a surrogate virus for disinfectant efficacy testing. MVA demonstrated the strongest predictive performance for FMDV, limited alignment with ASFV, and functioned as a conservative benchmark for AIV. Given the need for a unified, biosafety-compatible surrogate for field-level disinfection and regulatory testing, MVA provides a practical means to ensure baseline efficacy against diverse pathogens, particularly in settings without high-containment facilities.

## Figures and Tables

**Figure 1 viruses-17-01156-f001:**
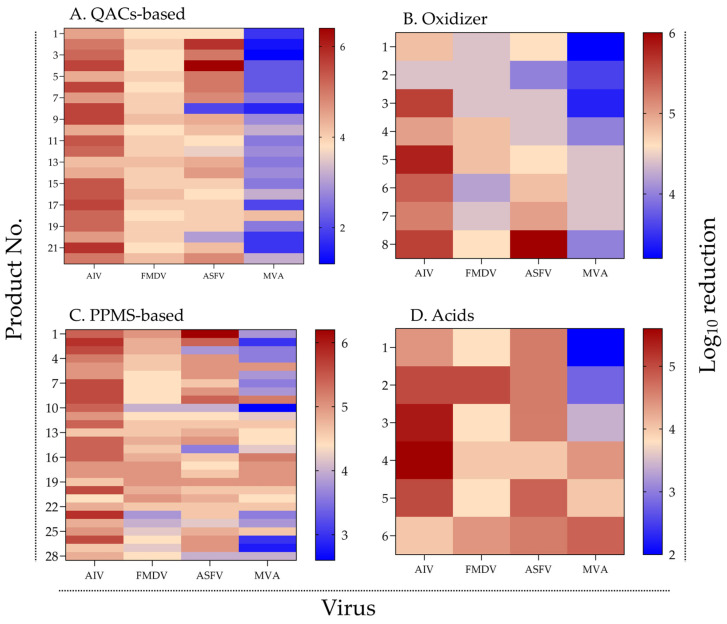
Virucidal efficacy of disinfectants by chemical group and virus type. Heatmaps showing log_10_ reduction in viral titers of AIV, FMDV, ASFV, and MVA after treatment with (**A**) QACs-based, (**B**) oxidizing agents, (**C**) PPMS-based, and (**D**) organic acid-based disinfectants. Color intensity indicates virucidal efficacy, with red representing high and blue representing low reduction. Warmer colors indicate higher virucidal activity (see Appendix A for raw data).

**Figure 2 viruses-17-01156-f002:**
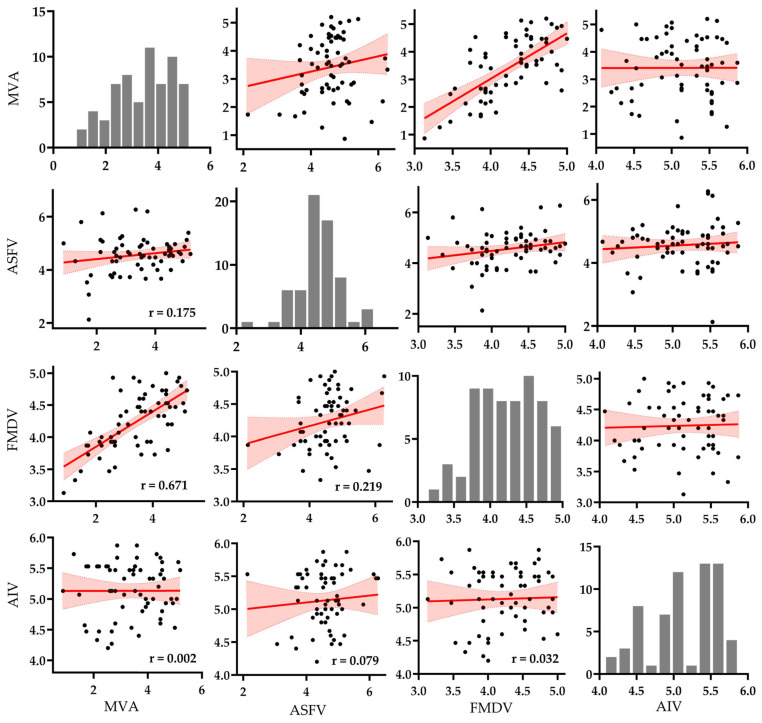
Pairwise scatter plot matrix of log_10_ reduction values (*n* = 64) among MVA, ASFV, FMDV, and AIV. Each off-diagonal panel displays a pairwise comparison with fitted linear regression lines (in red) and shaded 95% confidence intervals. Histograms along the diagonal show the distribution of log_10_ reduction values for each virus. Pearson correlation coefficients (r) are annotated in the lower triangle of the matrix (see Appendix A for raw data and analysis).

**Figure 3 viruses-17-01156-f003:**
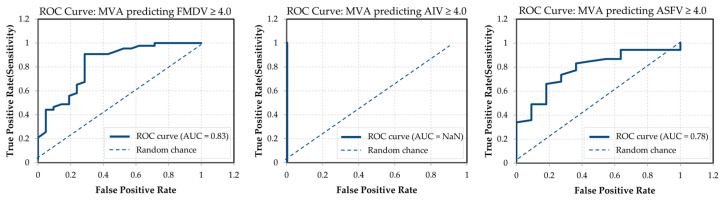
ROC curves for MVA-based prediction of disinfectant efficacy (≥4 log_10_ reduction). AUC values are shown in each panel. MVA exhibited strong classification performance for FMDV (AUC = 0.83) and ASFV (AUC = 0.78), while prediction for AIV was not possible due to insufficient variance in AIV efficacy data (raw data and analysis are provided in Appendix A).

**Figure 4 viruses-17-01156-f004:**
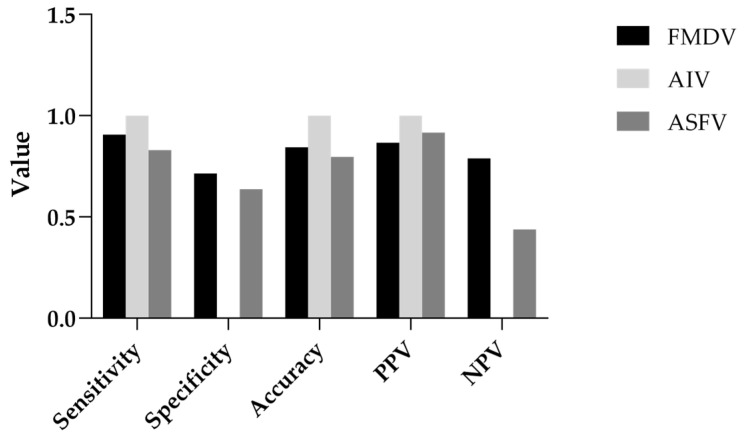
Classification metrics of MVA-based prediction for AIV, FMDV, and ASFV. Bar plots of classification metrics illustrating MVA’s predictive performance for disinfectant efficacy (≥4 log_10_ reduction) against AIV, FMDV, and ASFV. Metrics include sensitivity, specificity, accuracy, positive predictive value (PPV), and negative predictive value (NPV) calculated for each virus (see Appendix A for raw data).

**Figure 5 viruses-17-01156-f005:**
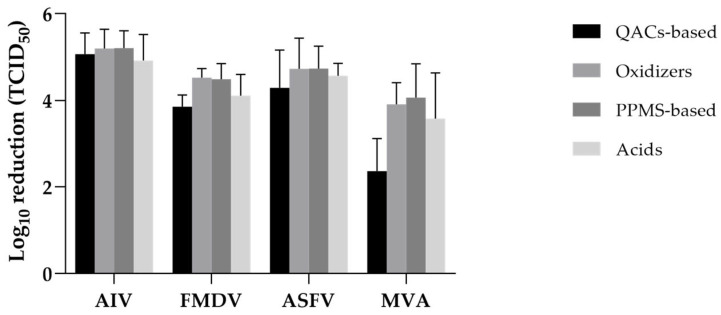
Mean virucidal efficacy of disinfectants by chemical group and virus. Bars represent mean values with standard deviation. PPMS-based and oxidizer groups generally achieved higher reductions across all virus types, whereas QACs-based products showed lower efficacy, especially against MVA (see Appendix A for raw data).

**Table 1 viruses-17-01156-t001:** Summary of Commercial Disinfectants by Active Ingredients.

Active Ingredient Type	Group	No. of Products	Example Composition
QACs-based	QACs + Organic Acids	21	BZK (100 g), CA (200 g), PA (60 g)
	QACs + Aldehyde	1	QACs (100 g), GLT (150 g)
Oxidizer	Peroxide-based Oxidizers	7	HP (250 g), PAA (50 g)
	Chlorine-based Oxidizers	1	NaDCC (3.3 g)
PPMS-based	PPMS + Organic Acids	23	PPMS (500 g), MA (100 g), SA (50 g)
	PPMS + Surfactant only	1	PPMS (500 g), SDBS (150 g)
	PPMS + Chlorine-based Oxidizer	2	PPMS (500 g), NaDCC (50 g)
	PPMS + Acid + Chlorine-based Oxidizer	2	PPMS (500 g), MA (150 g), NaDCC (51 g)
Acids	Organic Acids (Single or Mixed)	6	CA (200 g), AcOH (100 g), PA (100 g), Thymol (25 g)

BZK, benzalkonium chloride; CA, citric acid; PA, peracetic acid; MA, malic acid; PPMS, potassium peroxymonosulfate; SDBS, sodium dodecylbenzenesulfonate, QACs, quaternary ammonium compounds; GLT, glutaraldehyde; AcOH, acetic acid; SA, sulfamic acid; HP, hydrogen peroxide; PAA, peracetic acid; NaDCC, sodium dichloroisocyanurate. (The complete dataset of virucidal efficacy for all 64 disinfectants against AIV, FMDV, ASFV, and MVA is presented in Appendix A).

**Table 2 viruses-17-01156-t002:** Summary of virucidal efficacy by chemical group.

Virus	Group	Mean ± SD *	CV (%)	Efficacy (%) *
AIV	QACs-based	5.07 ± 0.49	9.7	99.9991
	Oxidizer	5.21 ± 0.44	8.5	99.9994
	PPMS-based	5.21 ± 0.40	7.7	99.9994
	Acids	4.92 ± 0.60	12.2	99.9988
FMDV	QACs-based	3.85 ± 0.27	7	99.9859
	Oxidizer	4.53 ± 0.21	4.6	99.9971
	PPMS-based	4.49 ± 0.36	8	99.9968
	Acids	4.11 ± 0.49	11.9	99.9923
ASFV	QACs-based	4.29 ± 0.88	20.5	99.9948
	Oxidizer	4.73 ± 0.71	15	99.9981
	PPMS-based	4.74 ± 0.52	11	99.9982
	Acids	4.57 ± 0.29	6.3	99.9973
MVA	QACs-based	2.37 ± 0.75	31.6	99.5705
	Oxidizer	3.91 ± 0.50	12.8	99.9876
	PPMS-based	4.06 ± 0.78	19.2	99.9914
	Acids	3.58 ± 1.06	29.6	99.9736

* Calculated from mean log_10_ reduction. Mean and SD values are derived from a panel of 64 commercially available disinfectants per virus. SD reflects inter-product variability, not experimental error.

**Table 3 viruses-17-01156-t003:** Strategic Use of MVA as a Surrogate for AIV, ASFV, and FMDV.

Parameter	AIV	ASFV	FMDV
Correlation (r)	0.002 (Negligible)	0.157 (Very low)	0.671 (Moderate)
AUC	Not applicable (Uniformly high efficacy)	0.78 (Fair)	0.83 (Good)
Quantitative Prediction	Not feasible	Limited	Feasible
Surrogate Utility	Conservative binary screening surrogate for highly sensitive viruses	Binary classification & Conservative screening	Binary classification & Conservative screening & Approximate quantitative prediction
Practical Application	Exclusion of ineffective disinfectants under conservative criteria	Screening and Exclusion of ineffective disinfectants	Pre-approval evaluation and potential replacement for live virus testing
Protective Threshold (MVA-kill implies full-kill)	Always (MVA-killing dilution always sufficient for AIV)	Often (but some borderline mismatches may occur)	Often (especially at conservative threshold of MVA ≤ 2.8)

## Data Availability

The data presented in this study are available in the article and its Appendix A.

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
