# Peer review of "Comparative Evaluation of Modified Vaccinia Ankara as a Surrogate Virus for Disinfectant Efficacy Testing Against AIV, FMDV, and ASFV"

_viruses, 2025, doi:10.3390/v17091156_

Round 1

Reviewer 1 Report

Comments and Suggestions for Authors

This study systematically evaluated the potential application of MVA as an alternative virus in the efficacy testing of disinfectants. It filled the gap in existing research regarding cross-virus type substitution validation. Regions lacking BSL-3 facilities provided feasible alternatives with clear practical value. However, the following points still need improvement:
1. In Section 2.3, please specify which virus was inoculated in the Vero cells.
2. Please explain in the Materials and Methods how many repetitions each experiment had.
3. The sample size of the Acid and Oxidizer groups was small, which may affect the universality of the statistical results.
4. The pH range of the disinfectant dilution solution was not clearly stated, and pH may affect the efficacy of certain disinfectants.
5. Further explanations are needed for the weak correlation between MVA and ASFV, such as whether it is related to the complexity of the virus envelope. It is suggested to conduct additional structural stability experiments for comparison, in order to enhance the research at the mechanism level. Such as the heat inactivation experiment.

Comments on the Quality of English Language

The English could be improved to more clearly express the research

Reviewer 2 Report

Comments and Suggestions for Authors

This well written manuscript describes studies to characterize the appropriateness of modified vaccinia Ankara as a surrogate, in inactivation studies, for three animal viruses of concern (avian influenza virus, foot and mouth disease virus, and African swine fever virus). Nice job on the preparation of the manuscript and the conduct of the studies. I just have a few suggestions for improvement of the manuscript.

Introduction, page 2, second paragraph. I would replace "highest dilution" with "highest concentration"

Introduction, page 2, third paragraph. What is meant by "HPAIV"? this is the first use of this abbreviation.

Materials and Methods. 2.1 some information about the LFBK cell line should be provided (e.g., animal species and tissue, source of the cells). Also the specifics of the antibiotic/antimycotic cocktail should be provided. The approximate speed and time for the centrifugation of the AI should be provided.

Materials and Methods. 2.3. For titer determination post neutralization, was the Vero cell line used for MVA, FMDV and ASFV? Or was the LFBK cell line used to titer one of these?

Table 1. I don't believe that SDBS has been defined anywhere.

Results 3.3. For the benefit of the readers you might explain what "ROC analysis" means

Page 10, first paragraph. Is there a reason why you show the SD for acid-based products instead of the CV?
